# HEM: An Improved Parametric Link Prediction Algorithm Based on Hybrid Network Evolution Mechanism

**DOI:** 10.3390/e25101416

**Published:** 2023-10-05

**Authors:** Dejing Ke, Jiansu Pu

**Affiliations:** 1Department of Computer Science and Engineering, University of Electronic Science and Technology of China, Chengdu 611731, China; dejingke@163.com; 2Big Data Visual Analysis Lab, University of Electronic Science and Technology of China, Chengdu 611731, China

**Keywords:** link prediction, complex networks, network evolution

## Abstract

Link prediction plays an important role in the research of complex networks. Its task is to predict missing links or possible new links in the future via existing information in the network. In recent years, many powerful link prediction algorithms have emerged, which have good results in prediction accuracy and interpretability. However, the existing research still cannot clearly point out the relationship between the characteristics of the network and the mechanism of link generation, and the predictability of complex networks with different features remains to be further analyzed. In view of this, this article proposes the corresponding link prediction indexes Reg, DFPA and LW on a regular network, scale-free network and small-world network, respectively, and studies their prediction properties on these three network models. At the same time, we propose a parametric hybrid index HEM and compare the prediction accuracies of HEM and many similarity-based indexes on real-world networks. The experimental results show that HEM performs better than other Birnbaum–Saunders. In addition, we study the factors that play a major role in the prediction of HEM and analyze their relationship with the characteristics of real-world networks. The results show that the predictive properties of factors are closely related to the features of networks.

## 1. Introduction

The network represents the relationship between entities in the form of connections, which is an effective and popular abstraction of the complex real world. Network science has been involved in biological, social, communication and economic fields and has had many fruitful achievements [1,2]. In the study of complex networks, the research on the formation and the evolution mechanism of networks has attracted more and more attention. These studies aim to understand the root causes of changes in network structure and function by studying the connection mode and the evolution rules of networks.

The research on the mechanism of network evolution has gone through a long process. The proposal of the Erds–Rényi model [3] realized the explanation of some of the randomness of the network, but the model cannot explain other characteristics of the network. Then, after the Watts–Strogatz (WS) small-world model [4] and the Barabási–Albert (BA) scale-free model [5] were proposed, the small-world and scale-free characteristics of complex networks were discovered, which led to important progress in the study of the network evolution mechanism. In the subsequent research on the network connection mechanism, there has been a fitness model [6], a local world model [7], a hierarchical structure model [8], a node replication model [9] and other models which also promoted the development of network science [10].

As an interesting and challenging research direction in network science, link prediction has attracted more and more attention. Link prediction aims to predict missing links and new links in the network through existing structural information in the network. At present, link prediction has been applied widely in recommendation systems [11,12,13,14,15], mining biological information [16,17,18,19], reconstructing network information [20,21], and evaluating network evolution models [22,23]. Moreover, link prediction also helps us to understand and infer the connection mechanism of complex networks. Link prediction is essentially a guess regarding the network evolution mechanism. A good link prediction algorithm can more accurately reveal the evolution behavior of a network [24]. Current link prediction methods mainly include methods based on structural similarity, network embedding, matrix completion, ensemble learning and neural network methods, etc. [13,25,26,27]. These methods calculate the connection probability between nodes in the network and express the network connection mechanism to some extent. Through the study of the network evolution mechanism, if we can deeply grasp the relationship between nodes in network evolution and deeply understand the basis of connections in the network, we are more likely to propose an excellent link prediction algorithm. Based on this idea, this article proposes a link prediction algorithm via the evolution characteristics of the network.

Among all the link prediction algorithms, the similarity-based algorithms are favored in many fields because of their simplicity and good interpretability. The link prediction algorithms based on structural similarity aim to predict links by calculating the similarity score between each pair of nodes. The higher the similarity score, the more likely the links are to be generated. Among the similarity Birnbaum–Saunders, the simplest and most intuitive index is the “common neighbor” (CN) [28,29]. CN assumes that the more common the neighbors between two nodes, the more likely they are to be connected. CN performs very well in the prediction on social networks. Many of the other local similarity Birnbaum–Saunders are based on common neighbors and taking into account the contribution of the degree of both ends of the nodes, such as Satlon, Jaccard, Sorensen, HPI, HDI, LHN1, etc. [28,30]. They are suitable for different network characteristics. In other local similarity Birnbaum–Saunders, AA and RA consider the influence of the degree of common neighbor nodes [28,31]. In recent years, some Birnbaum–Saunders that consider the local community paradigm (LCP) are integrated into the local similarity index [32], such as CAR, CRA and CH Birnbaum–Saunders [25,32,33]. These Birnbaum–Saunders believe that nodes within the community are more likely to have connections than nodes that are not in the same community. In addition to the local similarity Birnbaum–Saunders, there are some Birnbaum–Saunders that consider wider information, such as the LP index, which considers the information of higher-order paths, such as three-order and above [28,29]. Moreover, LRW and SRW calculate the similarity between nodes using random walk [26,34]. Similarly, global similarity Birnbaum–Saunders take into account information about the entire network, such as Katz, LHN2 and LO [29,30,35]. The more information is considered, the better the performance will be, but it also leads to a higher computational cost.

The research of link prediction and complex networks is developing rapidly, but it also faces many challenges. Firstly, the existing similarity algorithms often perform well in the face of a few networks, but they are no longer effective when dealing with a wider range of real-world networks, including directed networks, weighted networks, heterogeneous edge networks and other complex situations [36,37,38]. Secondly, there is a strong correlation between the link prediction algorithm, the network structure characteristics and the link predictability of the network in theory [39,40]. However, how to describe and express the relationship between them is a challenging task. In addition, through link prediction, the evolution characteristics of the network can be reproduced to a certain extent, and the research on the evolution behavior of complex networks can be promoted, but the research on this aspect is still relatively lacking; on the other hand, link prediction needs to face large-scale real data at the application level, and our algorithm needs stronger adaptability and more efficient calculations [41].

Therefore, starting from these challenges, this paper attempts to study through the following aspects. Firstly, this paper studies the characteristics of regular networks, scale-free networks and small-world networks. According to these characteristics, we propose the corresponding link prediction Birnbaum–Saunders Reg, DFPA and LW. Through these Birnbaum–Saunders, we aim to verify the following: link prediction Birnbaum–Saunders are often related to the characteristics of the network when predicting, a single index often cannot cope with many networks and Birnbaum–Saunders that fit a certain network characteristics will always be better for the network. After that, we propose a parametric hybrid index HEM. We hope that through this hybrid index, we can obtain a better generalization performance index that integrates the characteristics of different networks. This index has better adaptability and a more accurate prediction effect on complex real-world networks.

This article focuses on proposing link prediction algorithms according to the evolution mechanism of the network. Firstly, we experiment with the prediction ability of the link prediction Birnbaum–Saunders on some networks that are generated by some simple network evolution rules. Then, we experiment and compare the prediction accuracies of our hybrid index and some similarity-based Birnbaum–Saunders on some real-world networks.

The main contributions of this article can be summarized as follows:(1)This article proposes corresponding link prediction algorithms on regular networks, scale-free networks and small-world networks, respectively, and studies their prediction properties on these three network models.(2)This article proposes a parametric hybrid index, which has a higher prediction accuracy than many similarity-based Birnbaum–Saunders on real-world networks.(3)This article studies the main predictors in the hybrid index, and analyzes and summarizes their relationship with the network features.

In this article, we first introduce some basic network evolution models, then introduce the evaluation metrics of link prediction and some representative similarity-based algorithms. Finally, we introduce our proposed Birnbaum–Saunders based on the network evolution mechanism.

## 2. Network Model and Link Prediction

In this section, we will briefly introduce some network evolution models, link prediction evaluation metrics and similarity-based Birnbaum–Saunders.

### 2.1. Network Evolution Model

The study of complex networks plays an increasingly important role in mathematics, statistical physics, computer science and other fields [42,43,44,45,46]. Many biological, sociological and information systems can be represented by graphs, where nodes represent individuals and links represent relationships between individuals. Facts have proved that most complex networks are not random. Many complex networks have small-world property, scale-free property, community structure and other characteristics [4,5,47,48]. Therefore, many complex networks can be constructed using some simple rules.

In order to study the connection mechanism of networks, models such as the random graphs model, small-world model, scale-free model and hierarchical network model have been proposed [1,8,10]. These models can simulate the overall structure of the real network according to the potential network connection mechanism, so as to reproduce the observed attribute characteristics in the real network. This article will introduce the evolution characteristics of a regular network, small-world network and scale-free network.

#### 2.1.1. Regular Network

In the regular network, each node has the same number of neighbors, that is, the degree of each node is equal. Many crystal networks or protein networks in the field of chemistry can be regarded as regular networks.

#### 2.1.2. Scale-Free Network

Many observed networks have the characteristics of power–law degree distribution. In the power–law distribution, the degree of most of the nodes is very small, whereas the degree of a few of the nodes is very large. Networks with power–law distribution are also called scale-free networks [49]. For many complex networks in the real world, such as the Internet and social networks, they have the properties of scale-free networks.

At present, there are many mechanisms that can generate scale-free networks, such as the rich-get-richer mechanism [5], optimal design-driven mechanism [50], merging, regeneration mechanism [51] and so on [41]. In order to study scale-free networks, Albert-László Barabás and Réka Alber proposed a model for generating scale-free networks [5], which is the most classical BA model. The BA model explains the power–law distribution of the network through network growth and preferential attachment. The model is introduced below.

The BA model contains two important steps: growth and preferential attachment. Growth: new nodes are added to the initial network at each time point. Due to the addition of new nodes, the scale of the network is growing. Preferential attachment: when a new node is connected to the initial network, the probability of connection with the existing node is proportional to the degree of the node. Therefore, new nodes tend to be connected to nodes with a high degree. In the BA model, when a new node generates a connection, the probability of selecting a node with a degree of $k_{i}$ is:(1)$$P\left(i\right)=\frac{k_{i}}{\sum_{j}k_{j}}$$

The remarkable feature of the network generated by the BA model is the power–law distribution.

#### 2.1.3. Small-World Network

The small-world network depicts the phenomenon of the large clustering coefficient and the small average short path length in the real world network. The famous six-degrees of separation theory, Kevin Bacon game and Erdős number all reveal the universality of small-world characteristics in the real world [52,53]. Social networks, protein networks, food chain networks, cultural networks and so on have been proved to have the characteristics of small-world networks.

In the study of the mechanism of small-world networks, the WS model is a more classic small-world construction model [4]. The construction algorithm includes two steps:(1)Starting from the regular network, consider a regular network consisting of *N* nodes, circled in a ring. Each node is connected to its closest K/2 left neighbors and K/2 right neighbors in the ring, where *K* is an even number;(2)Randomly reconnect each link in the network with probability *p* (one endpoint of the link remains unchanged, while the other endpoint is a randomly selected node from the network). Note that there can be at most one link between any two different nodes, and each node cannot have a link connected to itself.

According to the WS model, p=0 corresponds to the regular network, and p=1 corresponds to the completely random network (see in Figure 1). Therefore, by adjusting the value of the *p*, we can control the transition from the regular network to the completely random graph. Network generated by WS model is a network between regular network and random network.

Similarly, the NW model is also a classic small-world network construction model [54]. The NW model simply changes the “randomized reconnecting” in the WS model to “randomized adding links”. The difference between the NW model and WS model is that the randomization process in the WS model may destroy the connectivity of the network, while NW does not cut off the original link in the regular network, but adds a link with probability *p*. The network constructed in this way has both a large clustering coefficient and small average distance.

### 2.2. Link Prediction Evaluation Metrics

Define G(V,E) as an undirected graph without multiple edges or self-connections, where *V* is the set of vertices and *E* is the set of edges. Define *U* as the universal set of edges, so U contains |V|(|V|−1)/2 edges. The task of link prediction is to find missing edges or possible edges in the future. These edges exist in the set U−E. Generally, link prediction algorithms are unsupervised learning methods and we usually do not know which edges need to be predicted.

In order to test the accuracy of the link prediction algorithm, it is common to divide the known edges *E* into training set $E^{T}$ and test set $E^{P}$, with $E^{T}\cup E^{P}=E$ and $E^{T}\cap E^{P}=\emptyset$. The edges in $E^{T}$ are the known information in the graph. Meanwhile, the edges in $E^{P}$ need to be predicted, which can be used to evaluate the accuracy of the link prediction algorithms. The $E^{T}$ and $E^{P}$ are randomly divided from the original data set. In order to overcome the statistical bias, this article uses the 10-fold cross-validation method [55,56].

Reference [26] proposed two methods to evaluate the accuracy of link prediction algorithms, namely AUC (area under the receiver operating characteristic curve) and Precision. They are introduced below.

AUC measures the accuracy of the algorithm by focusing on the whole link list, including the missing links (i.e., $E^{P}$) and the nonexistent links (i.e., U−E). Link prediction algorithms calculate the score of each pair of nodes. The greater the score the greater the probability of the existence of links between nodes. The AUC metric evaluates the accuracy of the algorithms by comparing the score of missing links and the nonexistent links.

Suppose there are n independent comparisons in total. Among these comparisons, there are $n_{1}$ times the missing link having a greater score and $n_{2}$ times the missing link and nonexistent link have the same score. Then, the AUC value can be calculated as:(2)$$AUC=\frac{n_{1}+0.5n_{2}}{n}$$

When AUC is equal to 0.5, the prediction accuracy of the algorithm is equivalent to that of a random prediction. The closer the AUC value is to 1, the better the prediction accuracy of the algorithm is.

The Precision metric sorts the scores of missing links and nonexistent links in descending order. We take the sorted top-*L* links as the predicted ones. Among these *L* links, *N* links belong to the test set. Then, the Precision can be calculated as:(3)$$Precision=\frac{N}{L}$$

Compared with AUC, the Precision only focuses on whether the top *L* links are predicted accurately.

### 2.3. Link Prediction Similarity-Based Algorithms

The similarity-based algorithms for link prediction compute a score $S_{xy}$ for each pair of nodes *x* and *y*, which is directly defined as the similarity between *x* and *y*. The algorithms consider that nodes with a high similarity score are more likely to connect. The similarity-based algorithms can be classified into two categories: local similarity Birnbaum–Saunders and global similarity Birnbaum–Saunders. Here, we choose some representative Birnbaum–Saunders to introduce (these Birnbaum–Saunders are similar to the Birnbaum–Saunders proposed in this paper in terms of expression, so they are chosen to better analyze and explain the differences. We ignore some Birnbaum–Saunders that are not comparable). The details are as follows.

#### 2.3.1. Local Similarity Birnbaum–Saunders


(1)Common Neighbor (CN) [28]:
(4)$$S_{xy}^{CN}=\left|\Gamma \left(x\right)\cap \Gamma \left(y\right)\right|$$


Γ(x) denotes the set of neighbors of the node *x*. In the CN index, the more common neighbors two nodes have, the more likely they are to connect.


(2)Salton Index [28]:
(5)$$S_{xy}^{Salton}=\frac{\left|\Gamma (x)\cap \Gamma (y)\right|}{\sqrt{k_{x}\times k_{y}}}$$


$k_{x}$ and $k_{y}$ denote the degree of nodes *x* and *y*, respectively. The Satlon index is also known as cosine similarity.


(3)Resource Allocation Index (RA) [28]:
(6)$$S_{xy}^{RA}=\sum_{z\in \Gamma (x)\cap \Gamma (y)}\frac{1}{k_{z}}$$


This index considers the dynamic resource allocation between *x* and *y* nodes. The resources that *x* sends to *y* are equally distributed among their common neighbors. So, the RA index defines the amount of resources *x* allocates to *y*, which represents the similarity score between the two nodes.


(4)Cannistraci–Hebb Index (CH) [33]:
(7)$$S_{xy}^{CH}=\sum_{z\in \Gamma (x)\cap \Gamma (y)}\frac{1+k_{z}^{i}}{1+k_{z}^{e}}$$
where $k_{z}^{i}$ denotes the number of links of *z* with other common neighbors of *x* and *y*, and $k_{z}^{e}$ denotes the number of links between *z* and nodes other than *x* and *y* or their common neighbors.



(5)Local Path Index (LP) [28]:
(8)$$S^{LP}=A^{2}+\epsilon A^{3}$$


ϵ is a free parameter. The LP index contains more information and performs better than the neighbor-based Birnbaum–Saunders. Obviously, when ϵ=0, this index degenerates to CN. LP index can be extended to include higher-order path information, as
(9)$$S^{LP(n)}=A^{2}+\epsilon A^{3}+\epsilon^{2}A^{4}+\ldots +\epsilon^{n-2}A^{n}$$
where *n* is the maximum order. The increase in n allows higher-order paths to be considered, but the computational complexity is also higher.

#### 2.3.2. Global Similarity Birnbaum–Saunders

(1)Katz Index [29]:

This index considers all path sets. It calculates all the paths and assigns less weight to long paths in an exponential decay. Katz can be expressed as:(10)$$S_{xy}^{Katz}=\beta A_{xy}+\beta^{2}A_{xy}^{2}+\beta^{3}A_{xy}^{3}+\ldots$$

β is the free parameter. The contribution of a higher order path can be controlled by adjusting β. Katz can also be written as
(11)$$S^{Katz}={\left(I-\beta A\right)}^{-1}-I$$

*I* is the identity matrix. β must be small enough to ensure convergence.

(2)Linear Optimization Index (LO) [35]:
(12)$$\begin{array}{cc} S^{LO} & =\alpha A{\left(\alpha A^{T}A+I\right)}^{-1}A^{T}A\\ \; & =\alpha A^{3}-\alpha^{2}A^{5}+\alpha^{3}A^{7}-\alpha^{4}A^{9}+\ldots \end{array}$$

α is a free parameter. *I* is the identity matrix and *A* is the adjacency matrix. When α is small enough, LO degenerates to the index that calculates only the three3-hop paths $A^{3}$.

## 3. Link Prediction Based on Network Evolution Mechanism

According to the characteristics of regular networks, scale-free networks and small-world networks, this article proposes link prediction Birnbaum–Saunders for these three networks, and proposes a hybrid index for complex networks based on the three Birnbaum–Saunders. Note that all the link prediction results in this article are obtained by using the 10-fold cross-validation method on test networks.

### 3.1. Index Based on Regular Networks

According to the characteristics of regular networks, this article proposes a link prediction index called Reg. Reg is expressed as follows:(13)$$S_{xy}^{Reg}=\frac{1}{\sqrt{k_{x}\times k_{y}}}$$

$k_{x}$ and $k_{y}$ represent the degree of nodes *x* and *y*, respectively. In the formula, the nodes with larger degree are less likely to be connected. Small nodes are more likely to generate connections. By suppressing the connection probability of large degree nodes and promoting the connection probability of small degree nodes, the degree balance is achieved to a certain extent.

In order to study the performance of the Reg index, we compared the link prediction accuracies of the Reg index, CN index and Salton index on a random regular network (see results in Table 1).

We can see that the Reg index is significantly better than other Birnbaum–Saunders. Due to the randomness of the regular network, the CN index has an AUC value of only 0.5, while the Satlon index shows random results even with the same computational factor (i.e., $\frac{1}{\sqrt{k_{x}\times k_{y}}}$) as the Reg index. As the degree of each node increases, the prediction performance of the Reg index will gradually decrease.

### 3.2. Index Based on Scale-Free Networks

In reference to the article [28], a link prediction index PA corresponding to the preferential attachment principle is proposed. The expression of PA is as follows.
(14)$$S_{xy}^{PA}=k_{x}\times k_{y}$$

This article also proposes a link prediction algorithm called DFPA (Difference Preferential Attachment) for scale-free networks. The expression is as follows.
(15)$$S_{xy}^{DFPA}=\frac{max(k_{x},k_{y})}{min(k_{x},k_{y})}$$

Compared with the PA index, the DFPA index pays more attention to the connection between nodes with a large degree and nodes with a small degree. Nodes with a similar degree are more stable and less likely to connect with each other. Therefore, small degree nodes and large degree nodes develop faster according to the DFPA index. Moreover, the connection probability between nodes with a large degree is smaller than PA.

We compare the link prediction accuracies of PA and DFPA on scale-free networks constructed by the BA model. The results are shown in Figure 2. Note that accuracies are measured by the AUC value. The number of nodes of the networks are all 2000. Based on the BA model, the new nodes generate 1, 2, 4, 8, 16, 32 and 64 links, respectively. Thus, there are seven kinds of scale-free networks.

According to the prediction results of PA and DFPA in these scale-free networks, DFPA performs better when the network is sparse. As the degree of each node increases, the performance of PA gradually becomes better, while that of DFPA shows a downward trend. However, DFPA has a higher upper limit than PA in prediction.

There is a definition of degree assortativity in article [57]: when it is greater than 0, nodes with similar degrees tend to connect with each other. When it is less than 0, nodes with different degrees are more likely to connect with each other. DFPA considers the latter case. In theory, the DFPA index also accurately predicts disassortative networks.

### 3.3. Index Based on Small-World Networks

In the first step of the WS model, each node is connected to the nearest *k* nodes. Based on that, this article proposes the LW (local world) index. The LW index considers that when two nodes have paths with a length less than *k* or k+1, it is possible for the two nodes to have a connection. The expression of the LW index is as follows.
(16)$$S^{LW}=A^{k}+A^{k+1}$$

*k* is the free parameter, *A* is the adjacency matrix of the network and $A^{k}$ calculates the number of paths with length *k* between each pair of nodes. The paths calculated by $A^{k}$ may go back and forth on some edges. For example, there are three cases that $A^{4}$ takes into consideration (see in Figure 3). In Case 1 and Case 2, there is a path of length 2 between *x* and *y*, but there is no path of length 4. The calculation of $A_{xy}^{4}$ includes round trips on some edges, including round trips on the direct path and on the branch. Here, we have $A_{xy}^{4}⩾A_{xy}^{2}>0$. In the third Case, there is only a path of length 4 between x and y, then $A_{xy}^{4}>0$ and $A_{xy}^{2}=0$. So, $A^{k}$ contains the information for $A^{k-2}$, $A^{k-4}\ldots$ In order to consider both odd-order paths and even-order paths, LW calculates the sum of $A^{k}$ and $A^{k+1}$.

*k* in the LW represents the breadth and scope of information, which is similar to *n* in the LP index. Compared with LP and Katz Birnbaum–Saunders, the LW index does not consider that the lower order path has a higher weight. The weight of the path is related to the size of *k* and the network structure. Moreover, the LW index has a small computational complexity.

To facilitate the comparison of the LP and LW Birnbaum–Saunders, we define the LPK index as:(17)$$S^{LPK}=A^{2}+A^{3}+\ldots +A^{k}+A^{k+1}$$

LPK is the case where the ϵ parameter of the LP is set to 1 and the order *n* of the LP is set to k+1. We define LP2, LP4 and LP8 as the cases where the *k* value of LPK takes 2, 4 and 8, respectively. Similarly, define LW2, LW4 and LW8 as the cases where the *k* value of the LW index takes 2, 4 and 8, respectively.

We select k=2,4,8, and then compare the prediction performance of CN, Katz, LPK and LW Birnbaum–Saunders on the networks generated by WS model (see result in Figure 4). Note that accuracies are measured according to the AUC value. The number of nodes of the networks are all 2000. According to the WS model, each node is set to connect to the nearest 8 and 16 neighbors and they are denoted as WS_8 and WS_16, respectively. The probabilities of link reconnection are set to 0.0, 0.25, 0.50 and 0.75 and they are denoted as r00, r25, r50 and r75, respectively. So, eight kinds of networks in total were generated here.

When p=0, all Birnbaum–Saunders perform very well, and the prediction accuracies basically reached 100%. On the whole, the more neighbors each node has, the better the performance of the Birnbaum–Saunders. With the increase in randomness, the prediction accuracies of all Birnbaum–Saunders decrease accordingly. It can be seen that the performance of LW2 and LP2, LW4 and LP4 and LW8 and LP8 are basically the same, which proves that compared to the LPK index, the LW does not lose much information on the link prediction in small-world networks.

### 3.4. Hybrid Index Based on Complex Network

Among the above three Birnbaum–Saunders, Reg and DFPA are Birnbaum–Saunders based on degree distribution, and LW is the index based on network topology. According to the three link prediction Birnbaum–Saunders proposed by different network models, this article proposes a hybrid index called HEM (Hybrid Evolution Mechanism). The expression of HEM is as follows.
(18)$$S_{xy}^{HEM}={S_{xy}^{Reg}}^{\alpha }\times {S_{xy}^{DFPA}}^{1-\alpha }\times S_{xy}^{LW}$$

According to Equations (Equation 13), (Equation 15) and (Equation 16), the above formula can be expanded as:(19)$$S_{xy}^{HEM}={\frac{1}{\sqrt{k_{x}\times k_{y}}}}^{\alpha }\times {\frac{max(k_{x},k_{y})}{min(k_{x},k_{y})}}^{1-\alpha }\times {\left(A^{k}+A^{k+1}\right)}_{xy}$$

There are two free parameters α and *k* in the HEM index. The α parameter is used to balance the degree distribution. The role of the *k* parameter is the same as in the LW, representing the range of paths included.

By adjusting the α parameter, we can achieve the optimal balance of the HEM index in the link prediction of the mixed networks of regular networks and disassortative networks. When α is close to 1, the HEM index tends to predict regular networks; when α is close to 0, the HEM index tends to predict disassortative networks. The *k* parameter represents the path range considered in the prediction of the LW index. If the *k* value is set too small, some high-order paths may not be taken into account for prediction. If it is too large, the paths that should not be considered will be involved. Therefore, the α and *k* parameters need to be adjusted simultaneously during the experiment.

In order to test the link prediction accuracy of the HEM index, this article selects the following network data sets (see in Table 2). The multiple edges are regarded as one single edge, and the directed edge is regarded as an undirected edge. The self-connections are not taken into account. In addition, we only consider the giant component when one network is not well connected.

PPI is a protein–protein interaction network [58]. NS is a network of co-authorships in the area of network science [59]. Grid contains information about the power grid of the Western States of the United States of America [4]. INT represents the router-level topology of the Internet [60]. PB is a network of hyperlinks between political blogs about politics in the United States of America [61]. Yeast is a protein–protein interaction network in budding yeast [62]. FB consists of “friends lists” from Facebook, whose data were collected from survey participants using the Facebook app [63]. HSS represents the network of friendships between users of the website hamsterster.com [64]. GrQc is the collaboration network from the e-print arXiv and covers scientific collaborations between authors whose papers are submitted to the General Relativity and Quantum Cosmology category [65]. AS is the network of autonomous systems of the Internet that are connected with each other [64,65]. ER is the international E-road network, a road network located mostly in Europe [66].

There are many similarity Birnbaum–Saunders in link prediction. This paper only selects some Birnbaum–Saunders that are similar to the Birnbaum–Saunders proposed in this paper in terms of expression. On the one hand, it is better to control variables and understand the factors that cause the difference in accuracies between indexes. On the other hand, some Birnbaum–Saunders are quite different from the indicators in this paper in terms of predictive properties and computational performance, so that the predictive differences of the indicators cannot be accurately grasped, and the interpretability is also poor.

So, this article compares the prediction accuracies of the HEM index and other similarity-based Birnbaum–Saunders such as CN, Salton, PA, RA, CH, LPK, Katz and LO on these networks. In these 11 networks, we calculate the AUC value and Precision value of these link prediction algorithms (see results in Table 3 and Table 4).

According to the results of AUC, the HEM performs much better than the other indexes in the Grid, INT, AS and ER networks. In the PPI, PB, Yeast, HSS, GrQc and NS networks, the prediction accuracies of HEM is also higher than other indexes. For the FB network, the HEM and many other indexes perform very well, and the prediction accuracies are basically reaching 100%.

According to the results of Precision, the performance of the HEM index in the PPI, FB and HSS networks is much better than in other indexes, especially in the PPI and FB networks, the Precision values of the HEM index are almost 1. The HEM also has a better improvement on the PB and GrQC networks compared to the classic indexes. In contrast, for the AUC results, the HEM index outperforms in the Grid, INT and AS networks, but underperforms in Precision compared to other indexes, which indicates that most of the correct predictions from the HEM index for these networks come from the second half of the lists of links.

Moreover, in the tables we see that the parameters of the HEM index differ when taking the maximum AUC and Precision values. Therefore, we need to study the role of parameters in the HEM index and their relationship with network characteristics.

## 4. Analysis of HEM Index

In order to understand the influence of different parameters, study which factor, including Reg, DFPA and LW, plays a major role in the prediction. Here, we propose two methods.

(1)Calculate the prediction accuracies of different factors separately, and choose two factors with the highest accuracies.(2)Sample α and *k*, then choose the top five combinations of the α and *k* parameters from where the HEM index has the highest prediction accuracy. The main influencing factors are determined by the chosen α and *k* parameters, where α takes the average value and *k* takes the mode.

The first method discusses the performance of individual factors and the second method calculates the parameters that have a greater impact on the prediction. In practical considerations, the second method is used as the main reference and the results obtained by the first method can help us have a better understanding of the characteristics of the network.

Here, we discuss the situation when the prediction accuracy is measured using the AUC value. The results of two methods may be different when measured by the Precision value, but they are performed in the same manner. In this article we consider five factors; they are Reg, DFBA, LW2, LW4 and LW8.

We calculate the prediction results of the individual factors on the above 11 networks using method 1 (see results in Table 5). Through method 2, we calculate the prediction results of the HEM index under different parameters and then choose the top five results and record the combinations of parameters. Among all the sampling parameters, the α values are 0.0, 0, 25, 0.5, 0.75 and 1.0, and the *k* values are 2, 4 and 8 (see results in Table 6).

As is known from the formula of the HEM index, when $\alpha^{*}$ is greater than 0.5, the impact of Reg is greater than that of DFPA and vice versa. When $\alpha^{*}$ is 0.5, it means that both factors have little effect on the prediction. So, in method 2, if $\alpha^{*}$ is equal to 0.5, we do not consider the Reg or DFPA factor. In method 1, if both maximum AUC values are produced by LW, only the LW factor with the largest AUC value is taken.

We compare the main factors of the 11 networks obtained by the two methods and the results are shown in Table 7.

It can be seen that the results obtained by the two methods are basically the same except for the three networks, Yeast, GrQc and NS. In Yeast, the main factors calculated by method 2 has DFPA, while in method 1, DFPA in Yeast performs better than Reg. In the GrQc and NS networks, the main factors obtained by method 2 have Reg, while according to method 1, the Reg factor performs worse than the DFPA factor. Therefore, the influencing factors cannot be simply determined by the individual prediction accuracy.

Observe the several networks with a high clustering coefficient: NS, FB, PB and GrQc. They have LW2 as their main factors based on the first method. LW2 performs very well on these networks, especially on FB. The FB network is a dense network, with a high clustering coefficient, and the prediction accuracies of the LW indexes basically reach 1. So, we guess that the LW index may be related to the clustering coefficient of the network. In addition, we can also observe that the density of the network has a certain influence on the prediction of LW. For example, although the NS network has the highest clustering coefficient, the average degree of the network is only 3.75, far sparser than the FB network, and the LW2 and LW4 indexes perform less well than on the FB network. Moreover, the main factors in the NS network obtained by the second method are Reg and LW4, indicating that due to the sparsity, a wider *k* in LW and additional consideration of regularity are needed to have a better prediction performance in the NS network. In addition, although the clustering coefficient of the HSS network is low, the network is denser and the performance of the LW index on the network is as good as that on the PPI and PB networks, whose clustering coefficient are much larger.

Both the Grid and ER networks are sparse, and the diameter of the two networks is very large compared to other networks. Therefore, the LW index needs to consider wider paths to predict the links. The main factors obtained in method 1 and method 2 are both LW8. The degree of assortativity of the INT and AS networks is observed to be negative, indicating that the networks have a tendency towards a differential connection. Thus, in these two networks, DFPA as their main factor performs the best among all the factors.

Moreover, the maximum degree of the network AS is 1485, indicating that the degree distribution is very unbalanced and the preferential attachment is more obvious. So, the prediction performance of the DFPA factor alone on the AS network is also better. The maximum degree ofthe GrQc, ER and NS networks is relatively small, indicating that the degree distribution of the network is relatively balanced. So, in these three networks, the corresponding results obtained in the second method indicate that Reg is their main factor. Although the Yeast network also has a small maximum degree, the degree assortativity is negative, indicating that connections on the network are still difference preferential. Correspondingly, in the second method, DFPA is the main factor in the Yeast network.

In summary, the Reg factor often acts on networks with relatively balanced degree distribution; that is, when the maximum degree is relatively small, we can take the Reg index into account to predict links. The DFPA index is usually more effective in networks with negative degree assortativity. The prediction performance of LW index is determined by clustering coefficient, average degree and network diameter. When clustering coefficient is higher and the network is denser, the link prediction of LW index is always more accurate. The size of the *k* of the LW index depends largely on the diameter and average distance of the network.

By arranging the above results, we compare the prediction results (measured using the AUC value) of individual factors and the hybrid index using tabular statistics (see in Table 8).

So, we can see that the main factor largely determines the upper limit of the prediction accuracy of the hybrid index.

In general, the hybrid index always has a better prediction performance than the single index. The prediction performance is mainly determined by the main factor, and other factors may have some influence to the prediction, which will help to improve the overall result.

If we can determine the factors that have a greater impact in the link prediction of different networks, then we can save the sampling of the parameters of the HEM index that have little impact and reduce the computational complexity. Depending on the upper limit of the main factors, we can also have some idea of the upper limit of the HEM index. Determining the main factors can also give us some insight into the characteristics of the network.

## 5. Conclusions and Future Work

The link prediction indexes in this article, including the prediction indexes for specific networks and the hybrid indexes for complex networks, are based on the idea of simulating the evolution mechanism of complex networks through simple rules. This method can simplify our research on the mechanism of network evolution, so as to understand the intrinsic properties of the network more deeply.

Firstly, this article constructs regular networks, scale-free networks and small-world networks and proposes our link prediction algorithms based on the properties of these networks. We perform link prediction on these networks to understand the advantages and limitations of different link prediction algorithms.

Secondly, we propose an algorithm that combines the above three indexes. The hybrid index sets two parameters for the prediction factors. In this article, we sample the parameters and perform predictions on some real-world networks. The results show that the hybrid index performs better in those complex networks than many classical similarity-based indexes.

Finally, we analyze the dominant factors of the hybrid index using two methods. Experiments show that the prediction results of the hybrid index are often determined by one or two main factors, and the upper limit of the prediction of the main factors often determines the upper limit of the prediction of the hybrid index to some extent. In addition, we find that the main factors in the real world network are always related to the characteristics of the network, which coincides with the prediction properties of the indexes proposed by different network models.

In the experiment, we see that the computational complexity of the hybrid index increases with the frequency of the parameter sampling. If we can use simple parameter sampling to outline the prediction accuracy curve of the hybrid index, it will greatly optimize our algorithm. When performing predictions, we often pay attention to the best results, and parameter sampling should also be oriented to the upper limit of the index. The two methods of determining the main factors proposed in this article solve the problem of parameter sampling and determining the upper limit to some extent. However, our goal is still to draw a prediction curve, through which we can better understand the predictive properties of the hybrid index.

In future work, we will further refine the link prediction algorithms according to the network evolution mechanism. Firstly, we will refine the algorithms based on degree distribution. This article only considers the mixed degree distribution of the regular network and the disassortativitive network. Therefore, in future research, it is necessary to consider the complete degree distribution to propose a more comprehensive link prediction algorithm. Secondly, we need to refine the link prediction algorithms based on topology structure. The algorithm in this article only considers the paths of specific range. In the face of more complex situations, we need more effective topology information to predict links.

## Figures and Tables

**Figure 1 entropy-25-01416-f001:**
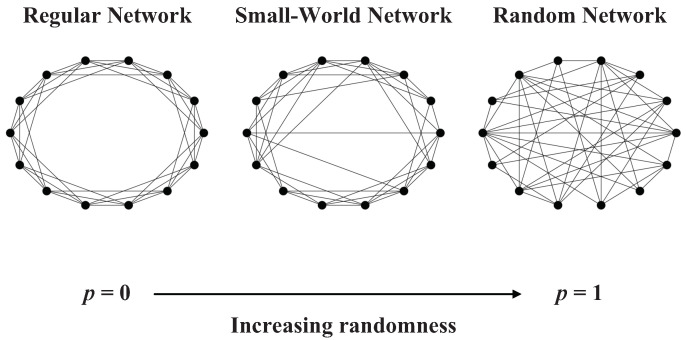
Illustration of the WS model.

**Figure 2 entropy-25-01416-f002:**
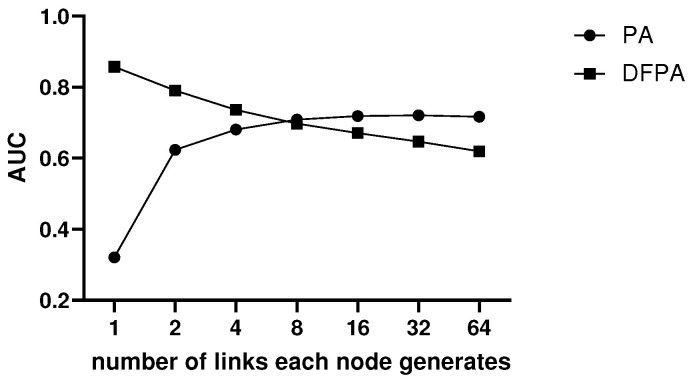
Accuracies of PA and DFPA on scale-free networks.

**Figure 3 entropy-25-01416-f003:**
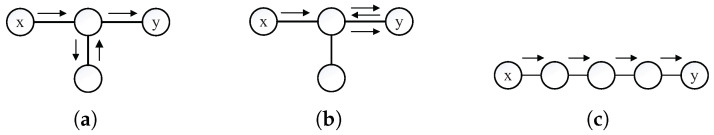
Example of the calculation of $A^{4}$. (**a**) Round trip on the branch; (**b**) round trip on direct path; (**c**) a four-order path.

**Figure 4 entropy-25-01416-f004:**
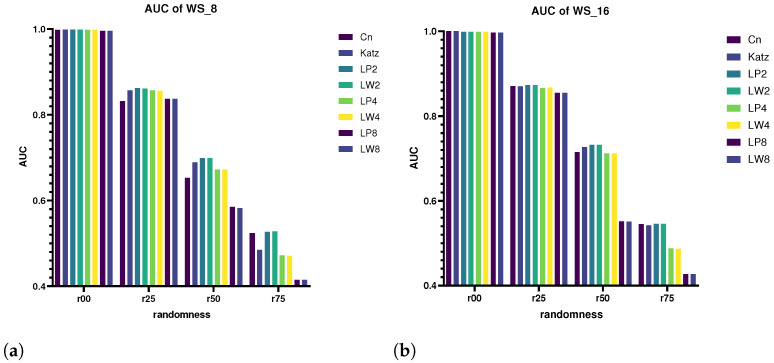
Accuracies of small-world networks with different randomness. (**a**) Accuracies of WS_8 networks; (**b**) accuracies of WS_16 networks.

**Table 1 entropy-25-01416-t001:** Accuracies of regular networks.

Network *	Reg_3	Reg_8	Reg_13	Reg_18	Reg_23	Reg_28	Reg_33
Cn	0.500	0.497	0.493	0.493	0.494	0.492	0.489
Salton	0.500	0.498	0.493	0.496	0.500	0.503	0.506
Reg	0.942	0.839	0.784	0.752	0.729	0.712	0.698

* Accuracies are measured using the AUC value. The number of nodes of the network are all 2000. The results are calculated using a random regular network where each node has 3, 8, 13, 18, 23, 28 and 33 neighbors, respectively. These seven regular networks are denoted as Reg_3, Reg_8, Reg_13, Reg_18, Reg_23, Reg_28 and Reg_33, respectively.

**Table 2 entropy-25-01416-t002:** The features of 11 real-world networks.

Network *	*N*	*M*	*K*	Δ	*D*	*C*	ρ
PPI	2375	11,693	9.85	118	15	0.306	0.454
NS	1461	2742	3.75	34	17	0.694	0.462
Grid	4941	6594	2.67	19	46	0.08	0.003
INT	5022	6258	2.49	106	15	0.012	−0.138
PB	1222	16,714	27.36	351	8	0.32	−0.221
Yeast	2361	6646	5.63	64	11	0.13	−0.099
FBC	4039	88,234	43.69	1045	8	0.606	0.064
HSS	1858	12,534	13.49	272	14	0.141	−0.085
GrQc	5242	14,484	5.53	81	17	0.53	0.659
AS	6474	12,572	3.88	1458	9	0.252	−0.182
ER	1174	1417	2.41	10	62	0.017	0.127

* *N* and *M* denote the number of nodes and edges of the network, respectively; *K* denotes the average degree; Δ denotes the maximum degree; *D* denotes the network diameter; *C* denotes the clustering coefficient; ρ denotes the degree of assortativity.

**Table 3 entropy-25-01416-t003:** The algorithms’ accuracies quantified using AUC.

Network *	PPI	Grid	INT	PB	Yeast	FB	HSS	GrQc	AS	NS	ER
Cn	0.893	0.589	0.559	0.919	0.706	0.992	0.805	0.922	0.696	0.943	0.526
Salton	0.892	0.588	0.559	0.875	0.705	0.992	0.789	0.922	0.676	0.944	0.526
PA	0.823	0.442	0.472	0.902	0.788	0.831	0.866	0.740	0.738	0.631	0.338
RA	0.894	0.589	0.559	0.923	0.706	0.995	0.809	0.923	0.700	0.944	0.526
CH	0.866	0.698	0.569	0.856	0.522	0.992	0.589	0.938	0.606	0.988	0.713
LP2	**0.939**	0.638	0.633	0.932	0.839	0.984	0.936	0.930	0.762	0.946	0.555
LP4	0.906	0.708	0.572	0.915	0.818	0.962	0.878	0.921	0.660	0.943	0.627
LP8	0.825	0.772	0.378	0.897	0.770	0.911	0.830	0.846	0.623	0.934	0.692
Katz	0.920	0.660	0.378	0.925	0.821	0.611	0.915	0.914	0.690	0.945	0.629
LO	0.935	0.560	0.623	0.929	0.813	0.986	0.952	0.846	0.787	0.852	0.486
** α **	0.50	0.75	0.50	0.75	0.00	1.00	0.75	1.00	0.00	1.00	1.00
** *k* **	*2*	*8*	*2*	*2*	*2*	*2*	*2*	*4*	*2*	*4*	*8*
**HEM**	**0.958**	**0.902**	**0.922**	**0.936**	**0.869**	**0.989**	**0.953**	**0.961**	**0.944**	**0.987**	**0.858**

* The parameter values in both Katz and LO Birnbaum–Saunders are set to 0.01. The values of the *k* parameter in LPK are selected as 2, 4 and 8, respectively. In the HEM index, we simultaneously sampled the α parameter and the *k* parameter. The values of α are selected as 0.0, 0, 25, 0.5, 0.75 and 1.0, respectively, and the values of *k* are selected as 2, 4 and 8, respectively. Among the 15 results obtained by combining the two parameters, we take the best result of the HEM index and record the α and *k* parameters when the AUC value is maximized.

**Table 4 entropy-25-01416-t004:** The algorithms’ accuracies quantified using Precision.

Network *	PPI	Grid	INT	PB	Yeast	FB	HSS	GrQc	AS	NS	ER
Cn	0.474	0.000	0.008	0.078	0.003	0.040	0.003	0.354	0.059	0.200	0.000
Salton	0.000	0.000	0.000	0.000	0.000	0.001	0.000	0.011	0.000	0.046	0.000
PA	0.409	0.000	0.014	0.082	0.009	0.033	0.089	0.222	0.131	0.005	0.000
RA	0.002	0.000	0.000	0.028	0.001	0.041	0.000	0.000	0.016	0.004	0.000
CH	0.267	0.005	0.000	0.010	0.008	0.006	0.000	0.140	0.026	0.229	0.000
LP2	0.548	0.037	0.280	0.412	0.144	0.661	0.297	0.629	0.253	0.252	0.000
LP4	0.531	0.046	0.243	0.391	0.117	0.689	0.186	0.641	0.227	0.253	0.000
LP8	0.523	0.035	0.218	0.349	0.099	0.694	0.161	0.644	0.213	0.251	0.001
Katz	0.533	0.001	0.009	0.261	0.003	0.612	0.015	0.522	0.099	0.201	0.000
LO	0.603	0.046	0.379	0.414	0.198	0.037	0.964	0.301	0.185	0.230	0.001
** α **	0.50	1.00	1.00	0.75	0.75	0.00	1.00	0.50	0.00	0.75	0.00
** *k* **	*2*	4	2	2	2	2	2	8	4	4	4
**HEM**	**0.978**	**0.051**	**0.159**	**0.524**	**0.178**	**0.993**	**0.731**	**0.759**	**0.081**	**0.273**	**0.002**

* The *L* value of Precision is 100. The parameter values in both Katz and LO Birnbaum–Saunders are set to 0.01. The values of the *k* parameter in LPK are selected as 2, 4 and 8, respectively. In the HEM index, we simultaneously sampled the α parameter and the *k* parameter. The values of α are selected as 0.0, 0, 25, 0.5, 0.75 and 1.0, respectively, and the values of *k* are selected as 2, 4 and 8, respectively. Among the 15 results obtained by combining the two parameters, we take the best result of the HEM index and record the α and *k* parameters when the Precision value is maximized.

**Table 5 entropy-25-01416-t005:** Accuracies of individual factors.

Network	Reg	DFBA	LW2	LW4	LW8
PPI	0.177	0.417	0.939	0.906	0.825
Grid	0.558	0.613	0.639	0.708	0.772
INT	0.528	0.849	0.633	0.571	0.378
PB	0.098	0.416	0.932	0.915	0.897
Yeast	0.213	0.580	0.839	0.818	0.770
FBC	0.169	0.261	0.984	0.962	0.911
HSS	0.134	0.525	0.936	0.878	0.830
GrQc	0.260	0.434	0.930	0.921	0.846
AS	0.262	0.929	0.762	0.660	0.622
NS	0.369	0.376	0.946	0.943	0.934
ER	**0.662**	0.565	0.555	0.627	**0.693**

**Table 6 entropy-25-01416-t006:** The top five combinations of α and *k*.

Network	$\alpha_{1}$	$k_{1}$	$\alpha_{2}$	$k_{2}$	$\alpha_{3}$	$k_{3}$	$\alpha_{4}$	$k_{4}$	$\alpha_{5}$	$k_{5}$	$\alpha^{*}$	$k^{*}$
PPI	0.50	2	0.75	2	1.00	2	0.25	2	0.00	2	0.5	2
Grid	0.75	8	1.00	8	0.50	8	0.25	8	0.00	8	0.5	8
INT	0.50	2	0.25	2	0.00	2	0.00	4	0.25	4	0.2	2
PB	0.75	2	1.00	2	0.50	2	0.25	2	0.00	2	0.5	2
Yeast	0.00	2	0.25	2	0.00	4	0.50	2	0.25	4	0.2	2
FB	1.00	2	0.75	2	0.50	2	0.25	2	0.00	2	0.5	2
HSS	0.75	2	1.00	2	0.50	2	0.25	2	0.00	2	0.5	2
GrQc	1.00	4	0.75	4	0.50	4	0.50	2	0.75	2	0.7	4
AS	0.00	2	0.25	2	0.50	2	0.75	2	0.00	4	0.3	2
NS	1.00	4	0.75	4	0.75	2	1.00	2	0.50	4	0.8	4
ER	1.00	8	0.75	8	0.75	4	1.00	4	0.50	8	0.8	8

$\alpha_{1}$–$\alpha_{5}$ and $k_{1}$–$k_{5}$ denote the five parameters with the best prediction performance, $\alpha^{*}$ denotes the average value of α and $k^{*}$ denotes the mode of k.

**Table 7 entropy-25-01416-t007:** The main factors of 11 networks obtained using method 1 and method 2.

Method	PPI	Grid	INT	PB	Yeast	FB	HSS	GrQc	AS	NS	ER
1	LW2	LW8	DFPA, LW2	LW2	LW2	LW2	LW2	LW2	DFPA, LW2	LW2	REG, LW8
2	LW2	LW8	DFPA, LW2	LW2	DFPA, NW2	LW2	LW2	REG, LW4	DFPA, LW2	REG, LW4	REG, LW8

**Table 8 entropy-25-01416-t008:** Results of individual factors and hybrid index.

	PPI	Grid	INT	PB	Yeast	FB	HSS	GrQc	AS	NS	ER
best of Factors	0.939	0.772	0.849	0.932	0.839	0.984	0.936	0.930	0.929	0.946	0.693
best of HEM	0.958	0.902	0.922	0.936	0.869	0.989	0.953	0.961	0.944	0.987	0.858

## Data Availability

Publicly available datasets were analyzed in this study. This data can be found here: http://www.konect.cc/networks/.

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
