# Peer review of "HEM: An Improved Parametric Link Prediction Algorithm Based on Hybrid Network Evolution Mechanism"

_entropy, 2023, doi:10.3390/e25101416_

Round 1

Reviewer 1 Report

The manuscript proposes link prediction algorithms on different networks and studies their prediction properties. It also analyzes different prediction indices and their connection to network features. 

The manuscript is an extension of an accepted conference paper. Basically, the authors have added some more details and evaluations and therefore it is acceptable as an extended version. The topic is relevant and interesting and the manuscript is well-structured and readable. Link prediction is an important issue in complex networks. The results support the conclusions and overall the manuscript can be published after maybe checking again for recent related publications in this research field.

The manuscript is well structured and readable.

Author Response

Dear Reviewers,

Thank you very much for your time involved in reviewing the manuscript and your very encouraging comments on the merits. We also appreciate your clear and detailed feedback and hope that the explanation has fully addressed all of your concerns.

Thank you for your praise of this article. And according to your suggestion of “checking again for recent related publications, we have checked over the related research articles carefully again. To best of our knowledge, we found that our article has taken into account the most recent relate publications to support our conclusions. Thank you for your kind suggestions.   

We would like to take this opportunity to thank you for all your time involved and this great opportunity for us to improve the manuscript. We hope you will find this revised version satisfactory.

Sincerely yours,   

Dejing Ke  

Reviewer 2 Report

1. Regarding Factor analysis (Table 5 )There are almost four networks in which they have cross loading issues.

2. The result in table 8 for comparing individual factors and hybrid index are very close to declare which one is performing better. Anova can be used to compare them based on significance differences.

Good Luck

Author Response

Thank you very much for your time involved in reviewing the manuscript and your very encouraging comments on the merits.

We also appreciate your clear and detailed feedback and hope that the explanation has fully addressed all of your concerns.

According to your first comment about the “cross loading issues in Table 5”, we have checked that thoroughly. The reason why we chose those four networks is the following: even though the four networks have the similar factors, they have different features (according the Table 2 in our article). Different features might cause the same factors. And our experiments were trying to find out how the network features relate to the specific factors.

And for your second suggestion about the “Anova can be used to compare them based on significance differences”. We have taken that in thorough consideration and we think that really helpful. In our experiments we try to find the main prediction factors and our conclusion is they are relating to the networks features. Therefore, we also think that maybe finding out the main factors by analyzing the network features might be more practical. Your kind advice is helpful for our future research.       

We would like to take this opportunity to thank you for all your time involved and this great opportunity for us to improve the manuscript. We hope you will find this revised version satisfactory.